# Maternal Metabolic State and Fetal Sex and Genotype Modulate Methylation of the Serotonin Receptor Type 2A Gene (*HTR2A*) in the Human Placenta

**DOI:** 10.3390/biomedicines10020467

**Published:** 2022-02-17

**Authors:** Marina Horvatiček, Maja Perić, Ivona Bečeheli, Marija Klasić, Maja Žutić, Maja Kesić, Gernot Desoye, Sandra Nakić Radoš, Marina Ivanišević, Dubravka Hranilovic, Jasminka Štefulj

**Affiliations:** 1Division of Molecular Biology, Ruđer Bošković Institute, HR-10000 Zagreb, Croatia; marina.horvaticek@irb.hr (M.H.); maja.peric@irb.hr (M.P.); ivona.beceheli@irb.hr (I.B.); maja.kesic@irb.hr (M.K.); 2Department of Biology, Faculty of Science, University of Zagreb, HR-10000 Zagreb, Croatia; marija.klasic@biol.pmf.hr (M.K.); dubravka.hranilovic@biol.pmf.hr (D.H.); 3Department of Psychology, Catholic University of Croatia, HR-10000 Zagreb, Croatia; maja.zutic@unicath.hr (M.Ž.); snrados@unicath.hr (S.N.R.); 4Department of Obstetrics and Gynecology, Medical University of Graz, A-8036 Graz, Austria; gernot.desoye@medunigraz.at; 5Department of Obstetrics and Gynecology, University Hospital Centre Zagreb, HR-10000 Zagreb, Croatia; marina.ivanisevic@pronatal.hr

**Keywords:** development, epigenetics, gestational diabetes mellitus, obesity, pregnancy, polymorphism, 5-HT, 5-HT2A

## Abstract

The serotonin receptor 2A gene (*HTR2A*) is a strong candidate for the fetal programming of future behavior and metabolism. Maternal obesity and gestational diabetes mellitus (GDM) have been associated with an increased risk of metabolic and psychological problems in offspring. We tested the hypothesis that maternal metabolic status affects methylation of *HTR2A* in the placenta. The prospective study included 199 pairs of mothers and healthy full-term newborns. Genomic DNA was extracted from feto-placental samples and analyzed for genotypes of two polymorphisms (rs6311, rs6306) and methylation of four cytosine residues (−1665, −1439, −1421, −1224) in the *HTR2A* promoter region. Placental *HTR2A* promoter methylation was higher in male than female placentas and depended on both rs6311 and rs6306 genotypes. A higher maternal pre-gestational body mass index (pBMI) and, to a lesser extent, diagnosis of GDM were associated with reduced *HTR2A* promoter methylation in female but not male placentas. Higher pBMI was associated with reduced methylation both directly and indirectly through increased GDM incidence. Tobacco use during pregnancy was associated with reduced *HTR2A* promoter methylation in male but not female placentas. The obtained results suggest that *HTR2A* is a sexually dimorphic epigenetic target of intrauterine exposures. The findings may contribute to a better understanding of the early developmental origins of neurobehavioral and metabolic disorders associated with altered HTR2A function.

## 1. Introduction

Overweight and obesity are becoming increasingly common worldwide, including among women of childbearing age [1,2]. Excessive body fat before pregnancy is associated with an increased risk of gestational diabetes mellitus (GDM), a condition characterized by impaired glucose tolerance first recognized during pregnancy [3]. Extensive epidemiologic data link both maternal obesity and GDM with increased offspring susceptibility to obesity, diabetes, and other metabolic disorders [4,5,6,7]. In addition, maternal obesity and GDM have been associated with an increased risk of neurodevelopmental and other mental health problems in offspring [8,9,10,11,12]. The molecular mechanisms underlying these associations are not yet clearly understood. However, they may involve epigenetic changes in the fetal genome that occur in response to an altered intrauterine environment [13,14]. DNA methylation, an epigenetic process in which methyl groups are covalently linked to cytosines, typically within CpG dinucleotides, is particularly sensitive to environmental influences during early development. It is a stable, mitotically inheritable DNA modification and is therefore considered a key mechanism linking an unfavorable prenatal environment with an increased risk of chronic physical and mental diseases in postnatal life [15].

The placenta plays a central role in regulating the fetal environment [16]. It not only controls the exchange of nutrients, gasses, and wastes between the maternal and fetal circulations, but is also an important source of hormones that regulate fetal development. In early pregnancy, for example, the placenta supplies the developing fetal brain with serotonin (5-hydroxytryptamine, 5-HT), a well-known monoamine neurotransmitter that acts as a potent trophic factor during development and regulates, among other (neuro)developmental processes, the growth of 5-HT neurons and the maturation of their target regions [17]. It has been recognized that disturbances in placental and fetal 5-HT homeostasis may contribute to the early developmental origins of behavioral disorders [18,19,20,21]. This might also be the case for metabolic disorders, as indicated by the influence of placental/fetal 5-HT level on later appetite regulation in a rat model [22]. Therefore, it is of utmost importance to better understand the pregnancy-related factors that contribute to individual variation in placental 5-HT homeostasis.

Homeostasis of 5-HT is maintained by several classes of 5-HT-related proteins, including transporters, metabolic enzymes, and receptors for 5-HT. Serotonin receptor 2A (HTR2A, also known as 5-HT2A) is one of the most widely distributed 5-HT-related proteins, found in both neuronal and non-neuronal tissues [23]. This G protein-coupled receptor plays a role in diverse physiological functions ranging from complex cognitive behaviors to vasoconstriction, platelet aggregation, immune response, and energy metabolism. Consistent with its broad physiological functions, HTR2A also plays a role in pathophysiological processes. It is of particular interest in biological psychiatry as a molecular substrate of emotional, neuropsychiatric, and neurodegenerative disorders [24,25] and an important target for psychotropic drugs [26,27]. In addition, HTR2A has been implicated in the pathophysiology of cardiovascular disorders [28,29], diabetes [30], and obesity [31].

HTR2A is encoded by a single gene (*HTR2A*) that spans approximately 66 kilobases on human chromosome 13q14-q21 [32]. Several common genetic variants in the *HTR2A* region, including rs6311, have been reported to regulate *HTR2A* expression [33,34,35], although results were not consistent [36,37]. In addition, transcription of *HTR2A* is modulated by methylation of several cytosines in the promoter region of the gene [34,38,39,40]. Changes in *HTR2A* methylation have been associated with various medical conditions, including schizophrenia and bipolar disorder [34,39,41], metabolic and obesity traits in subjects with metabolic syndrome [42], chronic fatigue syndrome [40], autism spectrum disorder (ASD) [43], substance use disorder [44], and multiple sclerosis [45].

Among many other tissues, *HTR2A* is also expressed in the placenta [46]. 5-HT signaling through HTR2A in the placenta has been implicated in placental development [47,48] and regulation of placental endocrine functions [49]. Interestingly, placental methylation of the *HTR2A* promoter region has been associated with neurobehavioral traits in healthy newborn infants [50], arguing for a role of *HTR2A* in the molecular mechanisms by which the prenatal environment influences neurodevelopmental trajectories [51].

Indeed, studies in rodents show that a low-protein [21] or high-fat [52] maternal diet during gestation has a lasting effect on *Htr2a* expression in the brain of the offspring. In addition, changes in brain *Htr2a* expression in response to stress in the early neonatal period have been documented and linked to methylation of the *Htr2a* promoter region [53]. In humans, changes in salivary *HTR2A* methylation have been associated with stress exposure and psychopathology in early childhood [54]. However, prenatal factors that contribute to the variation in *HTR2A* methylation in humans are largely unknown.

The aim of the present study was to investigate whether maternal obesity and GDM modulate *HTR2A* methylation in the placenta, a key organ involved in fetal programming and a relevant biomarker of the fetal environment. Given the sex dichotomy for incidence of metabolic and mental disorders and the sexually dimorphic programming in response to maternal obesity and GDM [55,56], we extended the study objective to include the possible influence of fetal sex on the association between maternal metabolism and placental *HTR2A* methylation.

## 2. Materials and Methods

### 2.1. Ethical Approval

Participants were part of an ongoing birth cohort study PlaNS (Placental and Neonatal Serotonin), established in Zagreb, Croatia, with the primary objective to investigate the impact of maternal obesity and GDM on placental and neonatal serotonin systems and offspring outcomes (project code: IP−2018-01-6547; 1 December 2018). The study was approved by the Ethics Committee of University Clinical Hospital Centre Zagreb (class: 8.1–18/162-2, number: 02/21 AG; approved 18 July 2018) and Bioethics Committee of Ruđer Bošković Institute, Zagreb (BEP-8761/2-2018; approved 26 November 2018). Women involved in the study provided written informed consent and were given a copy of it for personal record. All procedures were in line with the Declaration of Helsinki.

### 2.2. Participants

Pregnant women were recruited at the Department of Gynecology and Obstetrics, University Hospital Centre Zagreb. Inclusion criteria were absence of pre-existing diabetes and of known fetal anomalies or adverse pregnancy conditions other than GDM. Further, only pregnancies terminated by elective cesarean section were included. Demographic, anthropometric, obstetric, and clinical data were collected from medical records and questionnaires completed by mothers 1–3 days before delivery.

Pre-pregnancy body mass index (pBMI) was calculated from pre-pregnancy body weight and height, both of which were obtained from medical records and additionally confirmed by the participant. Based on pBMI, women were categorized as under-/normal weight (<25.0 kg/m^2^), overweight (25.0 to 29.9 kg/m^2^) and obese (>30.0 kg/m^2^). GDM diagnosis was based on the International Association of Diabetes and Pregnancy Study Groups (IADPSG) criteria [57] implemented in a Croatian clinical setting [58]. Gestational age was the number of weeks from the self-reported first day of the mother’s last menstrual period, except in several cases where it was adjusted according to established guidelines [59]. Smoking behavior was dichotomized into (1) never having smoked or having quit smoking at least 6 months before the start of the current pregnancy and (2) having smoked throughout pregnancy or having quit smoking during pregnancy; data that did not fit into these categories and ambiguous data were treated as missing data.

At the start of this study, the PlaNS cohort included 219 pairs of mothers and singleton newborns. We excluded mother–infant pairs with a birth weight of less than 2400 g (*n* = 5) or more than 4500 g (*n* = 6), a gestational age of less than 37 weeks (*n* = 3), a maternal pBMI of less than 16 kg/m^2^ (*n* = 1), a diagnosis of overt diabetes in pregnancy (*n* = 3), and no collected placental tissue samples (*n* = 2). Thus, a total of 199 mother–newborn pairs were included in the present analyses. None of the women included in the study received serotonin-targeting medications during pregnancy.

### 2.3. Placental Tissue Sampling and DNA Extraction

Tissue samples of the fetal part of the placenta were obtained within 5 min after delivery. Decidua was removed, and the standardized sampling procedure was used, as previously described in detail [60]. In brief, we excised and pooled tissue pieces from 10–12 positions in each placenta, 2–3 pieces per quadrant. Collected tissue samples were preserved in RNAlater RNA Stabilization Reagent (Qiagen, Hilden, Germany) according to the manufacturer’s recommendations and then stored at −80 °C until nucleic acid extraction. Genomic DNA was extracted from placental tissue using the GenElute Mammalian Genomic DNA Miniprep Kit (Sigma-Aldrich, St. Louis, MO, USA) according to the manufacturer’s protocol, including an optional RNase treatment step. The concentration and purity of isolated DNA was determined by spectrophotometry (NanoPhotometer^®^ N60/N50, Implen, Munich, Germany). All samples were checked for integrity by agarose gel electrophoresis and stored at −80 °C until further processing.

### 2.4. Bisulfite Pyrosequencing Quantification of HTR2A Promoter Methylation

Quantitative DNA methylation analysis targeted four cytosines in the *HTR2A* promoter region, located 1665 (L1), 1439 (L2), 1421 (L3), and 1224 (L4) bp upstream of the *HTR2A* start codon and corresponding to positions 4464, 4691, 4709, and 4905, respectively, in the NCBI reference sequence NG_013011.1 (i.e., positions 46897570, 46897344, 46897326, and 46897129, respectively, in GRCh38 chr13) (Figure 1). These loci were selected based on our previous findings [43] and the results of Paquette at al. [50].

Bisulfite conversion was performed with uniform amounts of DNA (800 ng per sample) using the EZ DNA Methylation-Gold Kit (Zymo Research, Irvine, CA, USA) following the manufacturer’s protocol. Bisulfite-converted DNA was amplified using the PyroMark PCR Kit (Qiagen, Hilden, Germany) according to the manufacturer’s instructions. Primer sequences for the analysis of L1 site were as described in our previous study [43]. Sequences of bisulfite PCR primers and two sequencing primers used for the analysis of the L2, L3, and L4 sites were as described in the study by Paquette at al. [50]. The specificity of all PCR products was verified by 2% agarose gel electrophoresis. Control PCR reactions using unconverted DNA as template did not yield detectable PCR products. Each PCR and quantitative pyrosequencing run included a negative control (without template) and a reference sample. Quantitative pyrosequencing used the PyroMark Q24 Advanced Pyrosequencing System and the PyroMark Q24 Advanced CpG Reagents (all from Qiagen, Hilden, Germany), according to manufacturer’s instructions. Methylation values that did not meet the pyrosequencing quality control and/or were identified as outliers were excluded from the analyses (5 samples at L1 loci, 2 samples at L2 loci, and 3 samples at L3 loci).

### 2.5. Genotyping of HTR2A Promoter Polymorphisms

Single nucleotide polymorphism (SNP) rs6311 (−1438 G > A) was genotyped by visual inspection of pyrograms resulting from the methylation assay for L2 and L3 sites, adjusted to account for the presence of both possible alleles [50]. The SNP rs6306 (−1421 C > T) was genotyped using the TaqMan SNP genotyping assay (assay ID: C_11696888_10, Thermo Fisher Scientific Inc., Foster City, CA, USA) according to manufacturer’s instructions. The genotypes obtained from placental DNA were consistent with those obtained from cord blood DNA of the same newborn.

### 2.6. Statistical Analyses

Data were tested for normal distribution using the D’Agostiono and Pearson normality test. Potential outliers were screened for using the ROUT method [61]. Continuous variables with normal distributions were compared using Student’s t test or one-way analysis of variance (ANOVA), both with Welch’s correction where appropriate, whereas those with non-normal distributions were compared using Mann–Whitney test or Kruskal–Wallis test, as appropriate. Chi-square (χ^2^) test or Fisher’s exact test (FET) were used to test for differences in frequency distributions. Correlations were analyzed using Pearson’s correlation coefficient (r_p_) for normally distributed continuous variables or Spearman’s rank correlation coefficient (r_s_) for non-parametric data.

The interaction between neonatal sex and maternal categorical metabolic variables on methylation was examined using two-way ANOVA with Sidak’s multiple comparisons test. In this analysis, maternal pre-gestational body weight status (pBWS) was dichotomized into under-/normal weight (pBMI < 25.0) and overweight/obese (pBMI ≥ 25). Glucose tolerance status (GTS) of the mother was categorized as normal glucose tolerance (NGT) and gestational diabetes mellitus (GDM).

A multiple linear regression model was used to determine whether maternal metabolic parameters pBMI and GTS were significant predictors of methylation levels in female and male placentas after adjusting for potential confounding variables. Suitability of variables for parametric analyses and collinearity between predictors were assessed before main analysis. The G*power program was used to assess the statistical power [62]. With a medium effect size, an α probability error of 5%, and a statistical power of 90% for a total of 6 predictors, the required minimum sample size was 88. Mediation effect of GDM on the relationship between pBMI and *HTR2A* methylation was assessed using the four-step procedure of Baron and Kenny [63].

Statistical analyses were performed with GraphPad Prism 8 (GraphPad Software Inc, San Diego, CA, USA) and IBM SPSS Statistics 21.0 for Windows (SPSS Statistics, Chicago, IL, USA). All statistical tests were two-sided and *p* < 0.05 was considered statistically significant.

## 3. Results

### 3.1. Maternal and Neonatal Characteristics

Characteristics of 199 mother–newborn pairs by sex of newborn are summarized in Table 1. Mothers of female (*n* = 92) and male (*n* = 107) newborns did not differ in metabolic parameters, i.e., pBMI, pBWS (under-/normal weight vs. overweight vs. obese), weight gain during pregnancy, family history of diabetes, glucose tolerance status (NGT vs. GDM), or in other recorded parameters (age, parity, nicotine and alcohol consumption during pregnancy) (Table 1).

All women diagnosed with GDM (*n* = 80) were treated with an adjusted diet. Some women also received oral antidiabetics (*n* = 11) or insulin (*n* = 4), and one woman both oral antidiabetics and insulin. Compared to women with NGT (*n* = 119), women with GDM had a higher pBMI (median 26.3 vs. 22.3 kg/m^2^, *p* < 0.0001) and a higher family history of diabetes (48.1% vs. 22.6%, *p* = 0.0003). Consistent with higher pBMI and adjusted diet, they also had lower weight gain during pregnancy (median 12 vs. 14 kg, *p* < 0.0001). Other maternal and neonatal characteristics were similar between the NGT and GDM groups (Table A1 in Appendix A).

In accordance with the study design, all newborns were healthy and born at term by elective cesarean section. Female and male newborns differed in body weight and body length, but not in ponderal index and gestational age at birth or genetic characteristics (Table 1). Genotype distributions of the rs6311 and rs6306 polymorphisms conformed to Hardy–Weinberg equilibrium in both the female and male groups (all *p* > 0.05, χ^2^ test). Only one homozygote of the rs6306 minor allele T was found among all participants.

### 3.2. Placental HTR2A Methylation in Relation to Fetal/Neonatal Sex and Genotype

The four analyzed cytosines in the *HTR2A* promoter region, located −1665 (L1), −1439 (L2), −1421 (L3) and −1224 (L4) bp upstream of the start codon, were methylated to varying degrees (*p* < 0.0001, Kruskal–Wallis test). Except for the L1 site (Figure 2a), the extent of methylation varied depending on the rs6311 or rs6306 genotype. Thus, an increasing number of the rs6311 minor allele A, which converts CpG to CpA at the L2 site, resulted in a sharp decrease in L2 methylation (Figure 2b). Similarly, the presence of the rs6306 minor allele T, which leads to the loss of cytosine available for methylation at L3, decreased L3 methylation (Figure 2c). In addition, the presence of the rs6311 minor allele A was associated with a slight but statistically significant increase in methylation at the L4 site (Figure 2d).

Methylation levels at L3 and L4 sites were highly correlated (Figure 2e). As the average methylation of these two sites (L3/L4 methylation) was associated with neonatal behavioral parameters in another study [50], we considered L3/L4 methylation in further analyses. L3/L4 methylation varied depending on the rs6306 genotype but not on the rs6311 genotype (Figure 2f).

L1, L3, L4 and L3/L4 methylation levels were statistically significantly higher in male than in female placentas (Table 1). This was true also for L2 methylation when comparisons were made between male and female placentas with the rs6311 GG genotype (54.4 ± 5.4 vs. 49.9 ± 6.0, *p* = 0.001) or rs6311 GA genotype (30.9 ± 4.1 vs. 28.7 ± 5.3, *p* = 0.032). Fetal/neonatal sex and genotype had an independent effect on methylation, i.e., no statistically significant interaction between them was found (all *p* > 0.05).

### 3.3. Placental HTR2A Methylation in Relation to Maternal Pre-Gestational Body Weight Status

We next investigated whether maternal pre-gestational body weight status (pBWS; pBMI < 25.0 vs. pBMI ≥ 25) could influence placental *HTR2A* methylation, considering a possible moderating effect of neonatal sex. Two-way ANOVA found a statistically significant interaction between neonatal sex and pBWS on L1 methylation; *post-hoc* testing showed that maternal L1 methylation was lower in maternal overweight/obesity (pBMI ≥ 25) only in female placentas (Figure 3a).

To account for the influence of the rs6306 polymorphism on L3/L4 methylation, analyses of L3/L4 methylation were performed separately in subjects stratified by rs6306 genotype. In the rs6306 CC subgroup (Figure 3b), we found a borderline significant (*p* = 0.054) interaction between neonatal sex and pBWS, with L3/L4 methylation statistically significantly decreased by maternal overweight/obesity only in female placentas. A similar pattern was seen in the rs6306 CT subgroup (Figure 3c), but only the main effect of sex was statistically significant, possibly due to the small number of subjects.

To account for the effect of the rs6311 polymorphism, analyses of L2 methylation were performed separately in subjects stratified by the rs6311 genotype. In the rs6311 GA subgroup (Figure 3d), we found a significant interaction between neonatal sex and pBWS, with L2 methylation lower in maternal overweight/obesity only in female placentas (Figure 3d). In the rs6311 GG subgroup, a similar pattern was seen, but only the main effect of neonatal sex was statistically significant (Figure 3e), while in the rs6311 AA subgroup (*n* = 35) no significant interaction or main effects of pBWS and neonatal sex were found (all *p* > 0.05).

### 3.4. Placental HTR2A Methylation in Relation to Maternal Glucose Tolerance Status

We also tested the interaction between maternal glucose tolerance status (GTS; NGT vs. GDM) and neonatal sex on placental *HTR2A* methylation. A significant interaction was found between neonatal sex and GTS on L1 methylation, with lower L1 methylation in GDM only in female placentas (Figure 4a). Both neonatal sex and GTS had significant main effects on L3/L4 methylation in the rs6306 CC subgroup (Figure 4b), while only the main effect of neonatal sex was significant in the rs6306 CT subgroup (*p* = 0.011). Only the main effect of neonatal sex on L2 methylation was statistically significant in the rs6311 GA (*p* = 0.034) and GG (*p* = 0.0006) subgroups, while no significant effects were found in the rs6311 AA subgroup (all *p* > 0.05).

### 3.5. Multiple Regression Analyses on L1 and L3/L4 Methylation in Male and Female Placentas

We next constructed a multiple linear regression model to determine whether maternal pBMI and GTS were significant predictors of L1 and L3/L4 methylation in female and male placentas after including potential confounding variables. Confounding variables were selected on the basis of statistically significant bivariate correlations with L1 or L3/L4 methylation in either the female, male, or total sample. Thus, a statistically significant positive correlation was observed between maternal weight gain during pregnancy and L1 methylation in female placentas (*p* = 0.048). In addition, tobacco use during pregnancy was associated with lower L1 (*p* = 0.014) and L3/L4 (*p* = 0.018) methylation in male placentas. Maternal age, parity, and alcohol consumption were not associated with L1 or L3/L4 methylation in the female, male, or total sample (all *p* > 0.05). The same was true for neonatal body weight, ponderal index, and gestational age (all *p* > 0.05). Due to L3/L4 methylation dependence on the rs6306 genotype (Figure 2f), the L3/L4 model was adjusted for the rs6306 polymorphism. In addition, we adjusted both the L1 and L3/L4 models for birth weight, as it differed between female and male newborns, but essentially the same results were obtained without this adjustment.

Regression analyses showed that in female placentas, both a higher pBMI and a GDM diagnosis were statistically significant predictors of lower L1 methylation (Table 2), whereas only a higher pBMI was a statistically significant predictor of lower L3/L4 methylation (Table 3). In male placentas, neither pBMI nor GTS were statistically significant predictors of L3/L4 methylation (Table 3), whereas the overall model with L1 methylation as an outcome was not statistically significant (Table 2). Interestingly, tobacco use during pregnancy predicted lower L1 methylation (although the overall model was not statistically significant) and lower L3/L4 methylation in male placentas. In female placentas, it was not a significant predictor of L1 or L3/L4 methylation. As expected, the rs6306 genotype CT was a significant predictor of lower L3/L4 methylation in both female and male placentas. Weight gain during pregnancy and birth weight were not significant predictors of methylation in either female or male placentas.

### 3.6. Mediation Analysis

Because both pBMI and GDM were significant predictors of L1 methylation in female placentas, we performed a mediation analysis in which we tested whether GDM was a mediator between pBMI and L1 methylation (Figure 5). The analysis showed that higher pBMI was associated with lower L1 methylation (path c) and a higher probability of GDM (path a), which in turn was associated with lower L1 methylation (path b). Together, pBMI and GDM explained more variance in L1 methylation (22.1%) than pBMI (14.1%) or GDM (14.7%) alone, implying that pBMI and GDM each contributed independently to L1 variance, with GDM acting as a partial mediator between pBMI and L1 methylation levels in female placentas.

## 4. Discussion

The present study is the first to demonstrate the influence of maternal obesity and GDM on *HTR2A* promoter methylation in the human placenta. The observed changes in the level of *HTR2A* promoter methylation may contribute to altered placental *HRT2A* gene expression and protein abundance. This may modulate placental 5-HT homeostasis and potentially affect long-term outcomes. The results also highlight the importance of biological factors, including fetal sex and genotype, in modulating placental *HTR2A* methylation.

Multiple studies in humans and animal models have shown that maternal obesity and GDM induce global and gene-specific DNA methylation changes in the placenta and other fetal tissues [13,14], which may contribute to the mechanisms underlying the developmental origins of health and disease [15]. We focused on the *HTR2A* gene in this study because it is a strong candidate for the fetal programming of future behavior and metabolism. During embryogenesis, HTR2A mediates the regulatory functions of 5-HT in the placenta [46,47,48,49] and the trophic functions of 5-HT in the developing fetal brain [64]. In adulthood, abnormal function of HTR2A has been associated with neuropsychiatric [24,25], cardiovascular [29], and metabolic diseases [30,31]. Finally, studies in animal models have directly demonstrated the role of *Htr2a* in mediating the effects of perinatal (in utero and neonatal) adversity on later behavior and metabolism [21,52,53].

We analysed four methylation sites in the *HTR2A* promoter region (Figure 1), which correspond to transcription factor binding sites and play a role in regulating *HTR2A* expression [34,38,39,40]. The extent of methylation at all four sites was higher in placentas of male than of female neonates. This observation is consistent with a recent study showing that most autosomal loci with sex-dependent DNA methylation in human term placenta are more highly methylated in male than in female placentas [65].

The neonatal *HTR2A* genotype was another important determinant of *HTR2A* promoter methylation in the placenta. In addition to the effect of a common SNP rs6311, which has been shown to impact *HTR2A* promoter methylation in various tissues [34,35,40,43,50,54], we found an apparent effect of a rare genetic variant rs6306. To our knowledge, the effect of rs6306 on *HTR2A* methylation has not been reported in other populations. It is not clear whether this rare genetic variant has been overlooked in other studies or whether it is extremely rare or absent in other populations, as is the case with the *HTR2A* variant rs76665058, which is found only in individuals of African ancestry [35]. In our present sample of 199 participants, we found the *n* = 1 homozygous genotype of the rs6306 minor allele T, while the overall frequency of the minor allele T was 8.8% (7.6% in the NGT and 10.6% in the GDM group, *p* = 0.367). In addition to the effect of rs6306 on *HTR2A* methylation reported here, our preliminary results (not shown) suggest a significant effect of rs6306 on *HTR2A* expression levels. These results argue for the consideration of rs6306 in future genetic and epigenetic association studies.

Of the four sites examined, L1 (−1665) was the most highly methylated and the only one not affected by either polymorphism. A previous study reported hypermethylation of the L1 site in the brain of individuals with schizophrenia [39], but to our knowledge there are no studies of this site in the placenta. We found that higher pBMI and GDM diagnosis of the mother were both associated with decreased L1 methylation in the placenta. Importantly, these associations were specific to female placentas, as shown by the statistically significant interactions of neonatal sex with pBWS (Figure 3a) and GTS (Figure 4a), and further supported by the multiple regression results in female and male placentas (Table 2). Considering that the regression analyses were performed in a subset of subjects (for whom data on maternal tobacco use were available), the consistency of the results supports the robustness of the moderating role of neonatal sex on the association between L1 methylation and maternal metabolism.

A simple mediation analysis performed in female placentas showed that GDM acted as a partial mediator of the effect of pBMI on L1 methylation. This suggests that excessive pBMI affects L1 methylation directly through the associated endocrine, inflammatory, and/or metabolic disturbances and indirectly by increasing the likelihood of GDM, with maternal hyperglycemia affecting L1 methylation in the same direction as the disturbances associated with excessive pBMI.

As mentioned above, the rs6311 polymorphism had a pronounced effect on L2 methylation due to loss of the CpG site in the presence of the rs6311 allele A. Due to the prevalence of the rs6311 polymorphism, L2 methylation was excluded from the analysis in a previous study of placental tissue [50]. Another study examined L2 methylation only in rs6311 GG homozygotes and found that it was increased in the brains of individuals with schizophrenia [34]. We found a significant interaction between neonatal sex and maternal pBWS on placental L2 methylation in rs6311 GA heterozygotes (Figure 3d) and a similar pattern in GG homozygotes (Figure 3e). Additional regression analyses in female and male carriers of at least one G allele (Appendix A, Table A3) supported a notion that methylation of this polymorphic CpG site might be affected not only by genotype but also by maternal pBWS in the same way as L1 methylation. However, a larger number of samples would be required to thoroughly investigate the influences on this site.

The mean methylation of L3 (−1421) and L4 (−1224) sites (L3/L4 methylation) in placentas from our study sample (39.1 ± 6.7%, *n* = 196 including the single rs6306 TT genotype) was comparable to that reported for term placentas in a sample of Caucasian and non-Caucasian participants (37.1 ± 8.8%, *n* = 444) [50]. Similar to L1 methylation, L3/L4 methylation was negatively associated with maternal pBMI in female placentas but not in male placentas (Figure 3b and Table 3). Regarding the effect of GTS on L3/L4 methylation, a statistically significant main effect of GTS on L3/L4 methylation was found by two-way ANOVA (Figure 4b), but GTS was not a statistically significant predictor of L3/L4 methylation in multiple regression analyses in female or male samples (Table 3), possibly because of insufficient statistical power.

In a previous study, L3/L4 methylation in the placenta was found to be related to neonatal attention and the quality of movement, neurobehavioral measures that may predict future attention, anxiety, and motor development [50]. In addition, alterations in L3 and/or L4 methylation in various tissues, including peripheral blood [40,43,44], saliva [54], and brain [28], have been linked to different disorders in infants and adults, ranging from neurodevelopmental disorders [43] and psychopathology in preschool children [54] to brain disorders with onset later in life [34,39,40,44].

A recent epigenome-wide study found an association between maternal pBMI and placental methylation of a number of CpGs for which the methylation in cord blood and peripheral blood in adolescent period was shown to be associated with maternal pBMI, or the methylation in peripheral blood and adipose tissue was shown to be associated with obesity traits in children and adults [66]. These findings suggest that changes in DNA methylation in the placenta in response to maternal pBMI correspond, in part, to changes in other fetal tissues and may be an indicator of adverse health outcomes in later life. It remains to be determined whether this is the case with the changes in *HTR2A* methylation observed here. Considering that most human studies have shown that female children are more susceptible to metabolic programming by maternal obesity and diabetes [55,67], the finding that *HTR2A* methylation in female placentas is more sensitive to maternal metabolic disturbances than in male placentas supports this possibility.

The molecular mechanisms by which maternal overweight and obesity affect placental methylation are not yet fully understood. There is evidence showing that altered metabolites may affect expression and activity of DNA-modifying enzymes reviewed in [68], underpinning the notion of obesity-related metabolic changes interacting with the epigenetic machinery.

Similar to altered maternal metabolism, tobacco use during pregnancy is also known to affect DNA methylation in the placenta [69]. Here, we found that tobacco use during pregnancy was associated with a statistically significant decrease in L3/L4 methylation in male but not female placentas. In a previous study in an undivided sample of female and male newborns, a marginally significant increase in L3/L4 methylation was observed in placentas exposed to maternal tobacco use [50]. Possible reasons for the different results could be differences in population structure and ethnicity, different criteria for categorizing tobacco use, and/or the sex-specific effect of tobacco on *HTR2A* methylation. To avoid a possible confounding effect of tobacco exposure during periconception, our non-smoking group included women who had quit smoking at least 6 months before the onset of pregnancy or had never smoked. Our finding of a reducing effect of tobacco exposure on placental *HTR2A* methylation is consistent with a negative correlation between years of tobacco use and *HTR2A* methylation in peripheral blood cells [44]. Furthermore, the finding of a statistically significant association between tobacco exposure and *HTR2A* methylation only in male placentas is consistent with the finding that tobacco use has a stronger impact on the DNA methylation clock in peripheral blood cells of male than female participants [70].

An important general observation of this study is that fetal sex stands out as a moderator of the association between intrauterine exposures (i.e., maternal metabolic disturbances, maternal tobacco use) and placental *HTR2A* methylation. The reason for the sex-specific response to the intrauterine environment may lay in sexually dimorphic development governed by a different feto-placental hormonal milieu [71]. Consistent with our observation are experimental studies in animal models showing a sex-specific role of *Htr2a* in mediating the effects of prenatal nutrition [47] and neonatal stress [53] on later metabolism and behavior. Taken together, these findings argue for separate analysis of females and males in studies examining how the early life environment influences biological systems.

Our study has several strengths. The placentas were from healthy, full-term infants who were all born by cesarean section, thus avoiding possible influences of the mode of delivery. Only elective cesarean deliveries were included to standardize the time interval between delivery and collection of placental tissue. All participants were from the same geographic area, which is inhabited by an ethnically homogeneous population of Southern Slavic (mainly Croatian) descent. The mothers of the female and male infants were similar in terms of metabolic parameters and lifestyle. We carefully considered the influence of neonatal sex and genotype. The major weaknesses are the lack of mechanisms behind the observed associations, the focus on only one gene, and the lack of investigation of an association between methylation changes and functional outcomes. The latter will be investigated in future follow-up studies of infants from the PlaNS birth cohort. Whether restricting the study to elective cesarean sections affects the representativeness of the cohort is not known and should be investigated further. As the study included newborns weighing between 2400 and 4500 g, the results may not be valid for a wider range of birth weights.

## 5. Conclusions

This is the first study to investigate the influence of an intrauterine environment on *HTR2A* methylation. We found that altered maternal metabolism, specifically increased pBMI and GDM, is associated with decreased *HTR2A* promoter methylation in female placentas, whereas tobacco use during pregnancy is associated with decreased *HTR2A* promoter methylation in male placentas. These results suggest that the *HTR2A* gene in the placenta is a sexually dimorphic epigenetic target of intrauterine exposures. Given the potential consequences of altered methylation in the placenta for the future health of the offspring, our results highlight the importance of timely changes in maternal lifestyle, even before conception. In addition, the results may contribute to a better understanding of the early developmental causes of neurobehavioral and metabolic disorders associated with altered HTR2A function.

## Figures and Tables

**Figure 1 biomedicines-10-00467-f001:**
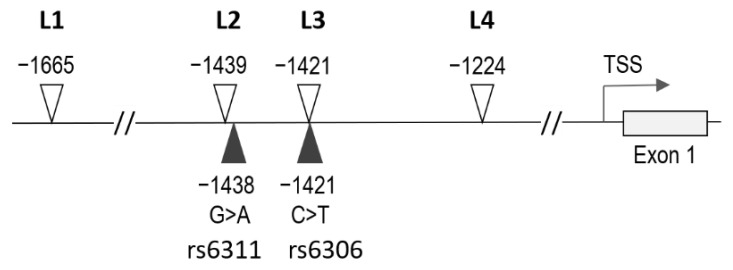
Schematic representation of the *HTR2A* promoter region. Positions of the analyzed methylation sites (L1, L2, L3, L4) are indicated by open triangles and those of the analyzed polymorphic sites (rs6311, rs6306) by closed triangles. Numbers refer to positions in bp relative to the ATG start codon in exon 1. The arrow indicates the major transcription start site (TSS).

**Figure 2 biomedicines-10-00467-f002:**
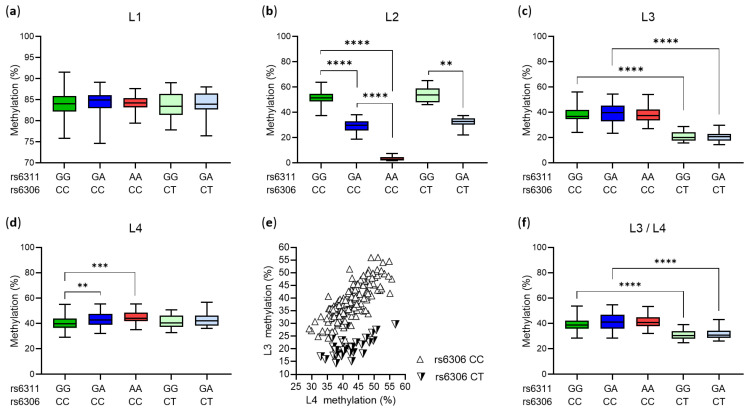
Placental *HTR2A* methylation in relation to neonatal genotype. (**a**–**f**) Subjects were divided into six subgroups based on the rs6311/rs6306 genotype combinations found in our study sample (GG/CC, *n* = 55; GA/CC, *n* = 73; AA/CC, *n* = 37; GG/CT, *n* = 20; GA/CT, *n* = 13; GG/TT, *n* = 1 (not shown)); methylation data are shown as boxplots with whiskers from minimum to maximum. Differences between subgroups were tested with Kruskal–Wallis test or one-way ANOVA: (**a**) *p* = 0.828; (**b**,**c**,**f**) *p* < 0.0001; (**d**) *p* = 0.0003. Where appropriate, multiple comparisons were made between rs6311/rs6306 genotype combinations differing by one of the two genotypes; significant differences are indicated (** *p* < 0.01, *** *p* < 0.001, **** *p* < 0.0001 by (**b**,**c**,**f**) Dunn’s or (**d**) Sidak’s *post-hoc* test). (**a**) Methylation at L1 (−1665) did not differ by rs6311 or rs6306 genotype. (**b**) Methylation at L2 (−1438) differed by rs6311 genotype. (**c**) Methylation at L3 (−1421) differed by rs6306 genotype. (**d**) Methylation at L4 (−1224) differed by rs6311 genotype. (**e**) Correlation between methylation levels at L3 and L4 sites in rs6306 CC homozygotes (r_p_ = 0.78, *p* < 0.0001) and rs6306 CT heterozygotes (r_p_ = 0.58, *p* < 0.0001). (**f**) Mean methylation at L3 and L4 sites differed by rs6306 genotype.

**Figure 3 biomedicines-10-00467-f003:**
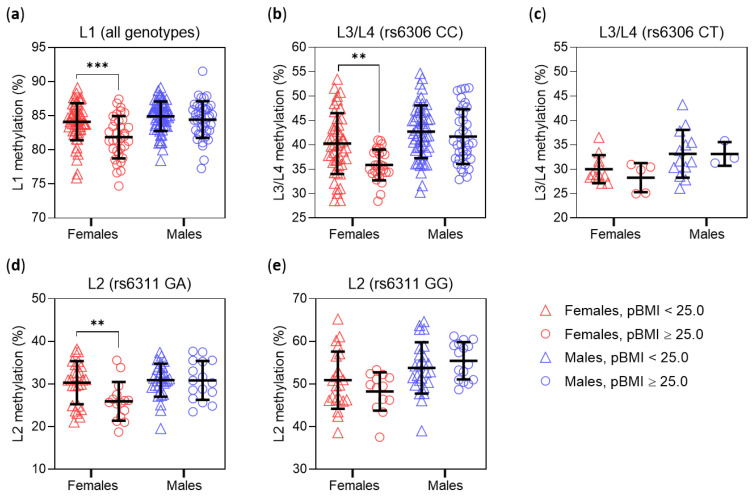
Placental *HTR2A* methylation as a function of neonatal sex and maternal pre-gestational body weight status (pBWS). pBWS was categorized based on maternal pre-gestational body mass index (pBMI). Individual values are shown, and lines indicate means and standard deviations. Statistical analyses used two-way ANOVA. (**a**) L1 methylation in the total sample (*n* = 194; *p* = 0.023 for interaction of pBWS and sex). L3/L4 methylation in (**b**) rs6306 CC homozygotes (*n* = 163; *p* = 0.054 for interaction of sex and pBWS, *p* < 0.0001 for main effect of sex, *p* = 0.003 for main effect of pBWS) and (**c**) rs6306 CT heterozygotes (*n* = 32; *p* = 0.020 for main effect of sex). L2 methylation in (**d**) rs6311 GA heterozygotes (*n* = 86; *p* = 0.036 for interaction of sex and pBWS) and (**e**) rs6311 GG homozygotes (*n* = 76; *p* = 0.0004 for main effect of sex). ** *p* < 0.01, *** *p* < 0.001 (Sidak’s *post-hoc* test).

**Figure 4 biomedicines-10-00467-f004:**
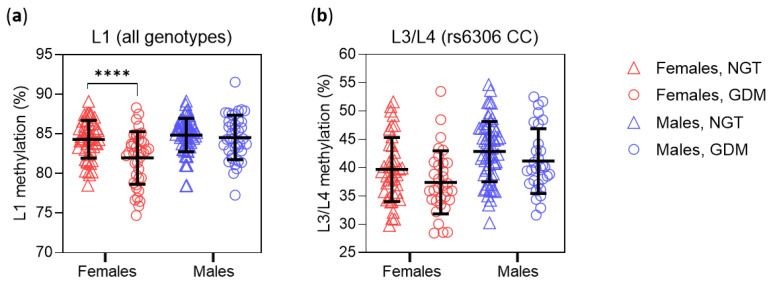
Placental *HTR2A* methylation as a function of neonatal sex and maternal glucose tolerance status (GTS). Individual values are shown, and lines indicate means and standard deviations. Statistical analyses used two-way ANOVA. (**a**) L1 methylation in the total sample (*n* = 194; *p* = 0.008 for interaction of GTS and sex). (**b**) L3/L4 methylation in rs6306 CC homozygotes (*n* = 163; *p* = 0.0002 for main effect of sex, *p* = 0.003 for main effect of GTS). NGT—normal glucose tolerance, GDM—gestational diabetes mellitus. **** *p* < 0.0001 (Sidak’s *post-hoc* test).

**Figure 5 biomedicines-10-00467-f005:**
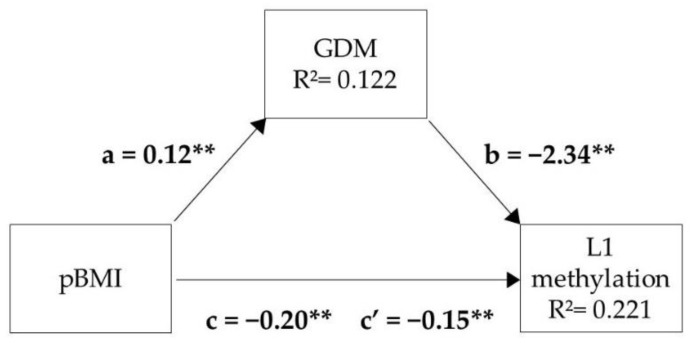
Simple mediation analysis model relating maternal pre-gestational body mass index (pBMI) and gestational diabetes mellitus (GDM) to *HTR2A* methylation at L1 site in female placentas (*n* = 89). The c path coefficient represents the effect of pBMI on L1 methylation. The c’ path coefficient refers to the effect of pBMI on L1 methylation, after controlling for the mediator GDM. R^2^ is the proportion of variance explained by the predictor variable in the regression model. Shown are unstandardized regression coefficients; all were statistically significant. For details, see Appendix A (Table A2). ** *p* < 0.01.

**Table 1 biomedicines-10-00467-t001:** Characteristics of the study sample by sex of newborn.

Characteristics	Females (*n* = 92)	Males (*n* = 107)	*p*-Value
** MOTHERS **			
Age at delivery, years	33.2 [30.1–36.9]	33.4 [29.7–36.7]	0.310 ^5^
Primipara/multipara, n (%)	40 (43.5)/52 (56.5)	35 (32.7)/72 (67.3)	0.143 ^6^
BMI before pregnancy, kg/m^2^	23.2 [21.4–27.7]	23.2 [21.0–28.4]	0.930 ^7^
BWS before pregnancy, n (%)			
normal weight	58 (63.0)	66 (61.7)	0.981 ^8^
overweight	15 (16.3)	18 (16.8)	
obese	19 (20.7)	23 (21.5)	
Diabetes in family (yes/no) ^1^	32 (35.2)/59 (64.8)	31 (30.7)/70 (69.3)	0.541 ^6^
NGT/GDM, n (%)	51 (55.4)/41 (44.6)	68 (63.5)/39 (36.5)	0.251 ^6^
Gestational weight gain, kg	12.5 [9.0–15.0]	14.0 [10.0–17.0]	0.102 ^7^
Tobacco in pregnancy (yes/no) ^2^	19 (22.1)/67 (77.9)	33 (32.7)/68 (67.3)	0.140 ^6^
Alcohol in pregnancy (yes/no) ^3^	16 (19.3)/67 (80.7)	12 (12.8)/82 (87.2)	0.303 ^6^
Oral antidiabetics (yes/no)	6 (6.5)/86 (93.5)	6 (5.6)/101 (94.4)	>0.999 ^6^
Insulin in pregnancy (yes/no)	1 (1.1)/91 (98.9)	4 (3.7)/103 (96.3)	0.376 ^6^
** NEWBORNS **			
Gestational age at birth, weeks	39.1 [38.5–39.6]	39.3 [38.7–39.7]	0.084 ^7^
Birth weight, g	3240 [3000–3615]	3520 [3290–3860]	**<0.0001** ^5^
Birth length, cm	49 [48–50]	50 [49–51]	**<0.0001** ^5^
Ponderal index, g/cm^3^	2.78 [2.62–2.99]	2.83 [2.65–2.97]	0.841 ^5^
rs6311 polymorphism, n (%)			
genotypes: GG/GA/AA	35 (38.0)/40 (43.5)/17 (18.5)	41 (38.3)/46 (43.0)/20 (18.7)	0.998 ^8^
alleles: G/A	110 (59.8)/74 (40.2)	128 (59.8)/86 (40.2)	>0.999 ^6^
rs6306 polymorphism, n (%)			
genotypes: CC/CT/TT	75 (81.5)/17 (18.5)/0 (0.0)	90 (84.1)/16 (15.0)/1 (0.9)	0.530 ^8,9^
alleles: C/T	167 (90.8)/17 (9.2)	196 (91.6)/18 (8.4)	0.860 ^6^
*HTR2A* methylation, % ^4^			
L1 (−1665 bp) site	83.8 [81.5–85.3]	85.0 [83.3–86.4]	**0.0006** ^7^
L2 (−1439 bp) site	32.7 [23.6–46.6]	33.9 [26.6–51.5]	0.212 ^7^
L3 (−1421 bp) site	34.5 [27.1–39.1]	37.9 [33.0–44.7]	**0.0005** ^5^
L4 (−1224 bp) site	40.3 [37.4–43.5]	44.0 [40.1–47.8]	**<0.0001** ^7^
Mean of L3 and L4 (L3/L4)	36.5 [32.2–40.8]	40.8 [36.1–45.2]	**<0.0001** ^7^

Continuous variables are reported as median [interquartile range] and categorical as number of subjects (n) and percentage (%). NGT, normal glucose tolerance; GDM, gestational diabetes mellitus; BMI, body mass index; BWS, body weight status. Statistically significant *p*-values are in bold. ^1−3^ Data are missing for *n* = ^1^ 7 (1 female, 6 males), ^2^ 12 (6 females, 6 males), and ^3^ 22 (9 females, 13 males) participants. ^4^ Methylation values not meeting pyrosequencing quality control and/or identified as outliers (including L3 methylation of *n* = 1 rs6306 TT genotype) were excluded and data shown correspond to *n* = 194 (89 females, 105 males) for L1, *n* = 197 (90 females, 107 males) for L2, *n* = 195 (89 females, 106 males) for L3 and L3/L4, and *n* = 199 (92 females, 107 males) for L4 methylation. ^5−8^ Comparisons between the female and male groups: ^5^ Student’s t test, ^6^ Fisher’s exact test, ^7^ Mann–Whitney test, ^8^ Chi-square test. ^9^ rs6306 genotype TT (*n* = 1) was not included in the calculation.

**Table 2 biomedicines-10-00467-t002:** Linear regression model of L1 methylation (%) predictors in female and male placentas.

Predictor	Female Placentas (*n* = 84)			Male Placentas (*n* = 99)		
	B	SE	*p*-Value	B	SE	*p*-Value
BMI before pregnancy (kg/m^2^)	−0.15	0.06	**0.010**	−0.06	0.04	0.181
Glucose tolerance status			**0.006**			0.979
NGT	ref.			ref.		
GDM	−1.93	0.68		−0.01	0.54	
Weight gain in pregnancy (kg)	−0.02	0.06	0.777	−0.01	0.04	0.765
Tobacco use in pregnancy			0.772			**0.024**
No	ref.			ref.		
Yes	−0.24	0.81		−1.24	0.54	
Birth weight (g)	−0.0005	0.0007	0.547	−0.0008	0.0006	0.248
R^2^ (adjusted R^2^)	0.240 (0.192)			0.084 (0.035)		
*p*	**0.0006**			0.141		

The unstandardized beta coefficient (B), standard error (SE) and *p*-values for each predictor in female and male placentas are presented. Statistically significant *p*-values are in bold. BMI—body mass index, NGT—normal glucose tolerance, GDM—gestational diabetes mellitus, ref.—reference.

**Table 3 biomedicines-10-00467-t003:** Linear regression model of L3/L4 methylation (%) predictors in female and male placentas.

Predictor	Female Placentas (*n* = 83)			Male Placentas (*n* = 100)		
	B	SE	*p*-Value	B	SE	*p*-Value
BMI before pregnancy (kg/m^2^)	−0.35	0.10	**0.001**	−0.03	0.09	0.725
Glucose tolerance status			0.442			0.386
NGT	ref.			ref.		
GDM	−0.94	0.68		−1.07	1.22	
Weight gain in pregnancy (kg)	−0.19	0.12	0.110	−0.05	0.09	0.591
Tobacco use in pregnancy			0.394			**0.023**
No	ref.			ref.		
Yes	−1.25	1.46		−2.79	1.21	
rs6306 polymorphism			**<0.0001**			**<0.0001**
CC genotype	ref.			ref.		
CT genotype	−9.65	1.44		−8.70	1.59	
Birth weight (g)	−0.002	0.001	0.261	−0.0002	0.0014	0.911
R^2^ (adjusted R^2^)	0.454 (0.411)			0.303 (0.258)		
*p*	**<0.0001**			**<0.0001**		

The unstandardized beta coefficient (B), standard error (SE) and *p*-values for each predictor in female and male placentas are presented. Statistically significant *p*-values are in bold. BMI—body mass index, NGT—normal glucose tolerance, GDM—gestational diabetes mellitus, ref.—reference.

## Data Availability

The data presented in the study are available from the corresponding author upon reasonable request.

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
