# Peer review of "Maternal Metabolic State and Fetal Sex and Genotype Modulate Methylation of the Serotonin Receptor Type 2A Gene (HTR2A) in the Human Placenta"

_biomedicines, 2022, doi:10.3390/biomedicines10020467_

Round 1

Reviewer 1 Report

The manuscript by Marina Horvatiček et al. investigated the effect of fetal sex and genotype and maternal metabolic state on the methylation of the HTR2A in human placenta. The study is of interest to the readers and I have the following suggestions:

1. have the authors checked the neonatal sex and genotype? The authors should discuss more about the potential effect of neonatal sex and genotype on the methylation of the HTR2A in human placenta.

2. How obesity changes the methelation? What are the potential mechanisms? This must be discussed. 

3. The authors should discuss more about the methylation changes and its potential outcomes. 

Reviewer 2 Report

There is an excellent study of high originality and very coprehensive results that can be published without any revision

Reviewer 3 Report

The work represents a valuable piece of information, although i would like to add that I have read this with a background on clinical research, with ‘limited lab analytical’ skills related to methylation assessment.

While the data do suggest association, supported by mechanistic causality for the neurodevelopment aspects, I still struggle with the link with metabolic disorders (lines 62-63). Is this secondary to the CNS issues ? Can the authors either provide more support for a potential causal relationship, or alternatively, soften this claimed causality as eg reference 41 discuss the difference in effectiveness of therapeutic interventions once metabolic syndrome is present.

Where women treated with serotonin ‘mediators’ ? excluded (SSRI or similar).

Although I understand the logistics related to elective caesarean intervention, can this affect the results, interpretation and subsequent extrapolation to the broader population ? the same holds true for elimination of ‘outlier’ weights (GDM mediated). You sell this as a strength, but perhaps some more reflection on this choice in the discussion is warranted

Was only one sample/placenta collected ? so no repeated measurements within the same placenta.

In your results section, you nicely discuss the consecutive findings (3.2, 3.3, 3.4, 3.5), but the mediation effort should perhaps better be reflected in the abstract ?

Was there any difference between primi- and multigravida women on the epigenetic findings ?
